# Combined BSA-Seq Based Mapping and RNA-Seq Profiling Reveal Candidate Genes Associated with Plant Architecture in *Brassica napus*

**DOI:** 10.3390/ijms23052472

**Published:** 2022-02-23

**Authors:** Shenhua Ye, Lei Yan, Xiaowei Ma, Yanping Chen, Lumei Wu, Tiantian Ma, Lun Zhao, Bin Yi, Chaozhi Ma, Jinxing Tu, Jinxiong Shen, Tingdong Fu, Jing Wen

**Affiliations:** National Center of Rapeseed Improvement in Wuhan, National Key Laboratory of Crop Genetic Improvement, College of Plant Science and Technology, Huazhong Agricultural University, Wuhan 430070, China; yeshenhua@webmail.hzau.edu.cn (S.Y.); yanlei@webmail.hzau.edu.cn (L.Y.); maxiaowei1224@163.com (X.M.); chenyanping9618@163.com (Y.C.); wulumei@webmail.hzau.edu.cn (L.W.); matian6887@163.com (T.M.); zhaolun@mail.hzau.edu.cn (L.Z.); yibin@mail.hzau.edu.cn (B.Y.); yuanbeauty@mail.hzau.edu.cn (C.M.); tujx@mail.hzau.edu.cn (J.T.); jxshen@mail.hzau.edu.cn (J.S.); futing@mail.hzau.edu.cn (T.F.)

**Keywords:** plant architecture, rapeseed, phytohormone profiling, BSA-Seq, RNA-Seq

## Abstract

Plant architecture involves important agronomic traits affecting crop yield, resistance to lodging, and fitness for mechanical harvesting in *Brassica napus*. Breeding high-yield varieties with plant architecture suitable for mechanical harvesting is the main goal of rapeseed breeders. Here, we report an accession of *B. napus* (4942C-5), which has a dwarf and compact plant architecture in contrast to cultivated varieties. A BC_8_ population was constructed by crossing a normal plant architecture line, 8008, with the recurrent parent 4942C-5. To investigate the molecular mechanisms underlying plant architecture, we performed phytohormone profiling, bulk segregant analysis sequencing (BSA-Seq), and RNA sequencing (RNA-Seq) in BC_8_ plants with contrasting plant architecture. Genetic analysis indicated the plant architecture traits of 4942C-5 were recessive traits controlled by multiple genes. The content of auxin (IAA), gibberellin (GA), and abscisic acid (ABA) differed significantly between plants with contrasting plant architecture in the BC_8_ population. Based on BSA-Seq analysis, we identified five candidate intervals on chromosome A01, namely those of 0 to 6.33 Mb, 6.45 to 6.48 Mb, 6.51 to 6.53 Mb, 6.77 to 6.79 Mb, and 7 to 7.01 Mb regions. The RNA-Seq analysis revealed a total of 4378 differentially expressed genes (DEGs), of which 2801 were up-regulated and 1577 were down-regulated. There, further analysis showed that genes involved in plant hormone biosynthesis and signal transduction, cell structure, and the phenylpropanoid pathway might play a pivotal role in the morphogenesis of plant architecture. Association analysis of BSA-Seq and RNA-Seq suggested that seven DEGs involved in plant hormone signal transduction and a WUSCHEL-related homeobox (WOX) gene (*BnaA01g01910D*) might be candidate genes responsible for the dwarf and compact phenotype in 4942C-5. These findings provide a foundation for elucidating the mechanisms underlying rapeseed plant architecture and should contribute to breed new varieties suitable for mechanization.

## 1. Introduction

Plant architecture is defined as the three-dimensional organization of plant organs, including plant height, branch or tiller angle, and number, leaf shape, and other organs’ morphology [1,2]. Plant architecture directly affects the adaptability of species to cultivation, the potential yield of crops, and their suitability to modern agricultural production operations, such as mechanical harvesting. Plant height is recognized as a fundamental trait of plant architecture that affect lodging resistance and crop yield. After the “Green Revolution”, the importance of dwarf and semi-dwarf crop varieties became widely known. Several key genes associated with plant height and lodging resistance have been cloned and shorter varieties with improved lodging resistance have been developed by introducing the dwarfing genes [3,4]. The optimum plant height, tiller angle, and branch numbers jointly determine photosynthetic efficiency and planting density and therefore the potential biomass yield increase of a crop. Many plant-architecture-related mutants in rice, wheat, maize, and *Arabidopsis* have been identified and their molecular mechanisms were revealed to assist in breeding new varieties with an ideal plant phenotype [2].

Phytohormones are key regulators in the modification of plant architecture. The genes involved in IAA, brassinosteroids (BRs), and strigolactone (SL) biosynthetic and signaling pathways, play important roles in controlling plant height, shoot branching, and tiller angle. For example, overexpression or gain-of-function of indole-3-acetic acid amido synthetase (*GH3*), an auxin-responsive protein in rice, reduced levels of IAA and eventually led to enlarged leaf angles and a dwarf phenotype [5,6,7]. In *Arabidopsis*, a frame-shift mutation (*GAI*) in GA 20-oxidase resulted in a dwarf phenotype insensitive to GAs, demonstrating the role of GA signaling in governing plant height. The orthologs of *GAI* in wheat (*Rht*) and maize (*D8*) are perhaps the most famous “Green Revolution” genes [3,8]. Recent research has revealed that jasmonic acid (JA) is also correlated with plant height determination. For instance, high levels of JA inhibited stem growth in *Nicotiana attenuata* by repressing the expression of both *GA20ox* and *GA13ox* in GA biosynthesis [9]. Phytohormone biosynthetic pathways have been described in detail and much evidence suggests there is cooperation and crosstalk between the signaling pathways of different hormones during plant growth [10]. Moreover, cell-wall-related genes, lignin synthetic genes, and homeobox genes are also implicated in determining plant architecture. In rice, a single nucleotide substitution of the *cellulose synthase-like D4* (*DNL1*) gene generated dwarfism and produced narrower leaves [11]. Overexpression of *PcBRU1*, a gene for the xyloglucan endotransglucosylase/hydrolases (XTHs) family, loosened the cell wall, leading to greater plant height and internode length in tobacco [12]. Phenylalanine ammonia-lyase (PAL) is a key enzyme in phenylpropanoid and lignin biosynthesis, and from the up-regulation of *PAL* arose a dwarf phenotype in *Ricinus communis* L. [13]. In rice, overexpressing the WUSCHEL-related homeobox gene *OsWOX3A* resulted in a reduced content of bioactive GAs and a dwarf phenotype [14].

Rapeseed is among the most important crops cultivated worldwide because it provides edible oil. Currently, China is the second-largest producer of rapeseed, but an advancement of its production there is hindered by low economic benefit limits. Accordingly, it is imperative that new rapeseed varieties suitable for mechanical harvesting are cultivated. Thus, exploring new mutants with desirable plant architectural traits and understanding their underlying genetic basis are the issues focused upon by rapeseed genetics and breeding in China. Currently, those plant architecture mutants reported in *B. napus* are mainly dwarf mutants related to GA or IAA biosynthesis and signaling. DELLA proteins act as GA signaling repressors in plants, and amino acid substitutions to DELLA proteins can affect how they interact with a gibberellin receptor, GID1, thereby changing the GA signaling and resulting in dwarfism in *B. napus* [15,16]. Both *BnaA3.IAA7* and *BnaC05.IAA7* (auxin/indole-3-acetic acid protein 7) were identified as key genes in dwarf mutants of *B. napus* [17,18,19,20]. Disrupting the conserved motif of these two genes changed auxin signaling, resulting in plant height changing from tall to dwarf. In the past two decades, a few quantitative trait loci (QTL) and candidate genes for branching-related traits were identified, but none have yet to be functionally validated [21,22,23,24,25]. Hence, our knowledge of the molecular mechanism underlying plant architecture in *B. napus* remains limited and awaits exploration.

In this study, we report on a *B. napus* line (4942C-5) whose plant architecture contrasts starkly with that of cultivated varieties. A high-generation segregating population with a genetic background of 4942C-5 was constructed by using 4942C-5 and a normal plant architecture line as the parents. Phytohormone profiling in this population was performed using a liquid chromatography tandem mass spectrometry (LC-MS/MS) system. Next, genes related to dwarfism and a compact canopy were identified via combined BSA-Seq and RNA-Seq analyses. The results presented here can provide a strong foundation for uncovering mechanisms responsible for rapeseed plant architectural traits and for breeding new rapeseed varieties apt for mechanized agriculture.

## 2. Results

### 2.1. Phenotypic Characteristics of 4942C-5

The *B. napus* line 8008 features a normal plant architecture, with a plant height of 136.57 ± 1.71 cm, main inflorescence length of 52.14 ± 2.50 cm, primary branch number of 6.07 ± 0.64, and branch length of 52.50 ± 2.58 cm. Compared with 8008, the plant height (76.94 ± 12.02 cm) and main inflorescence length (16.00 ± 2.12 cm) of 4942C-5 was significantly shorter at the mature stage (Figure 1). The primary branch number of 4942C-5 (6.75 ± 1.03) was not significantly different from that of 8008. However, the average length of branches of 4942C-5 (9.49 ± 0.67 cm) was only 18.08% that of 8008. Additionally, 4942C-5 plants formed a compact canopy with smaller branch angles. The plant architecture of F_1_ plants was similar to that of 8008; that is, neither plant height nor average branch length was significantly different between 8008 and the F_1_ plants (Figure 1). The frequency distributions of plant height, the main inflorescence length, and the average length of primary branches in the BC_8_ population each followed a normal distribution (Appendix A). These results suggest that the plant architecture traits of 4942C-5 were recessive traits controlled by multiple genes. Among the offspring of crosses between 4942C-5 and 8008, we found several lines whose plant architecture was deemed suitable for mechanical harvesting, namely semi-dwarf plant height, narrow branch angle, and short silique layer (Appendix A). Therefore, the mutant 4942C-5 provides a promising source of germplasm for breeding new rapeseed varieties whose plant architecture would be ideal for modern agricultural production.

### 2.2. Identification of Phytohormones

To investigate the relationship between phytohormones and plant architecture, the content of IAA, GA, ABA, SL, cytokinin (CK), ethylene (ETH), and BRs were each determined in tall and dwarf plants in the BC_8_ population by LC-MS/MS (Figure 2, Appendix A). Except for SL, ETH, and ABA, a total of 29 types of phytohormones were identified, including four forms of GAs, eight forms of CK, 12 forms of auxin, and five forms of BRs. Compared with tall plants, the content of gibberellin A9, N-(3-indolylacetyl)-L-valine, indole-3-carboxylic acid, indole-3-carboxaldehyde, indole-3-acetic acid, 3-indoleacetonitrile, 3-indole acetamide, typhasterol, and ABA was significantly increased in dwarf plants. In contrast, the gibberellin A15, gibberellin A20, N6-isopentenyl-adenine-7-glucoside, castasterone, and brassinolide content was significantly decreased in dwarf plants. Next, the total concentration of each class of phytohormones was calculated. These results showed that both the tall and dwarf plants accumulated much IAA, whose content was more than 2500 ng/g in the samples. Conversely, for GA, SL, CK, and BRs, their respective content in both plant architecture was extremely low, being less than 15 ng/g in the samples. For SL, CK, ETH, and BRs, no significant differences were detectable between the tall and dwarf plants, but the content of IAA, GA, and ABA was significantly different between them. Remarkably, the content of IAA was 1.41- to 1.66-fold higher in dwarf than tall plants, whereas the former’s GA or ABA content was significantly lower than the latter’s. Altogether, these results indicated that phytohormones, primarily IAA, GA, and ABA, play important roles in the determination of plant architecture in dwarf plants.

### 2.3. BSA-Seq Analysis

To rapidly map the candidate genes associated with the plant architecture traits of 4942C-5, BSA-Seq analysis was performed. The DNA from tall and dwarf plants was bulked in an equal ratio to generate a tall pool and a dwarf pool, respectively. These two pools, as well as the DNA extracted from the parents, were subjected to whole-genome resequencing. After filtering the raw data, 165.66 Gb of clean data were obtained having a Q30 ratio ≥ 92.13% and GC content  ≥38.07%. The mapped reads amounted to 170,027,343 for the tall and 209,301,252 for the dwarf pool, accounting for 90.99% and 90.74% of their total reads, and covering 46× and 57× of the assembled reference genome, respectively (Appendix A). Overall, 1,647,873 SNPs and 182,352 SNPs were identified between the parents and the two pools, respectively. After fitting a loess regression, the plant architecture genes were found to be related to five candidate intervals on chromosome A01, these located at 0–6.33 Mb, 6.45–6.48 Mb, 6.51–6.53 Mb, 6.77–6.79 Mb, and 7–7.01 Mb (Figure 3), with a total of 1256 genes annotated in the *B. napus* reference genome of “Darmor-bzh”.

### 2.4. Transcriptome Sequencing Data Analysis

To elucidate the molecular mechanisms underlying the phenotypic differences between dwarf and tall *B. napus* plants, six cDNA libraries with three biological replicates were sequenced. After filtering out any low-quality sequences and adaptors, a clean data set totaling 37.82 Gb was obtained having a Q30 ratio ≥93.23%. The mapping rate varied from 90.94% to 91.09% (Appendix A). Twelve DEGs in the RNA-Seq data were selected for validation using qRT-PCR. These results showed that the relative expression levels of DEGs by qRT-PCR corresponded well to the RNA-Seq data (R^2^ = 0.8764; Appendix A), verifying the reliability of the transcriptome data.

### 2.5. Expression Pattern Analysis of DEGs

A total of 4378 DEGs were identified between the tall and dwarf plants, of which 2801 were up-regulated and 1577 down-regulated (Figure 4A). To investigate trends in the expression of these DEGs in tall and dwarf plants, they were classified into six clusters by the K-means clustering algorithm method (Figure 4B) as follows: 1376, 787, 790, 585, 711, and 129 genes in cluster 1, cluster 2, cluster 3, cluster 4, cluster 5, and cluster 6, respectively. Overall expression levels of DEGs in cluster 3 were the highest among all clusters. In clusters 1, 2, and 3, the DEGs expression was up-regulated in the dwarf plants vis-à-vis tall plants. In contrast, expression levels of DEGs in clusters 4, 5, and 6 were all lower in dwarf plants than tall plants. Interestingly, 60.26% of 1296 DEGs in clusters 4 and 5 are located on chromosome A01. The results show that these DEGs are characterized by a complex expression patterning and might be relevant to the formation of plant architecture in *B. napus*.

### 2.6. Functional Analysis of DEGs

To understand the function of the DEGs, their GO enrichment analysis was performed. The top-50 enriched terms were divided into three ontology categories, i.e., molecular function, cellular component, and biological process (Figure 5A). Specifically, just a single GO term, vacuole, was enriched under the cellular component. Under the biological process, however, 39 GO terms were identified, many of these related to plant hormone biosynthesis and signal transduction, including response to abscisic acid, response to jasmonic acid, response to salicylic acid, cellular response to hormone stimulus, hormone-mediated signaling pathway, salicylic acid-mediated signaling pathway, response to auxin, and hormone biosynthetic process. Several GO terms related to enzyme activities were found enriched under molecular function, including cysteine-type endopeptidase activity, galactosidase activity, long-chain-fatty-acyl-CoA reductase activity, and glucosidase activity. A KEGG pathway analysis was also conducted to determine the possible roles of the DEGs in biological metabolic pathways. The DEGs were significantly enriched in 17 KEGG pathways (Figure 5B). As expected, plant hormone signal transduction was the most enriched pathway, implicating its vital role in the regulation of plant architecture in *B. napus*. The other 16 enriched pathways were mainly related to pathological reactions, amino acid metabolism, and metabolite synthesis, including plant–pathogen interaction, MAPK signaling pathway-plant, tyrosine metabolism, tryptophan metabolism, starch and sucrose metabolism, ubiquinone and other terpenoid-quinone biosynthesis, phenylpropanoid biosynthesis, phenylalanine metabolism, flavonoid biosynthesis, carotenoid biosynthesis, amino sugar and nucleotide sugar metabolism, linoleic acid metabolism, fatty acid degradation, glyoxylate and dicarboxylate metabolism, alpha-linolenic acid metabolism, and the AGE-RAGE signaling pathway in diabetic complications.

### 2.7. DEGs Involved in Plant Hormone Biosynthesis

The LC-MS/MS results had shown that the content of IAA, GA, and ABA in dwarf plants differed significantly from those in tall plants. Therefore, DEGs related to IAA, GA, and ABA biosynthesis were investigated (Appendix A). In the IAA biosynthesis pathway, expression of a *TAA1* gene encoding a tryptophan aminotransferase-related protein decreased by 3.03-fold in dwarf plants. By contrast, a *SURA1* gene encoding a S-alkyl-thiohydroximate lyase and another encoding a nitrilase were significantly up-regulated in dwarf plants. As expected, expression levels of two genes for gibberellin 20-oxidase were significantly lower in the dwarf than tall plants, consistent with the lower GA content in dwarf plants. Among the 10 DEGs related to ABA biosynthesis, only two genes (*BnaA09g51010D*: beta-carotene isomerase and *BnaAnng27240D*: (+)-abscisic acid 8′-hydroxylase) were down-regulated significantly in dwarf plants. Noticeably, alpha-linolenic acid metabolism, which provides precursors for JA biosynthesis, was enriched in the KEGG analysis. Most DEGs involved in alpha-linolenic acid metabolism were up-regulated in dwarf plants, including *lipoxygenase 2*, *allene oxide synthase*, and *allene oxide cyclase 2* (Appendix A).

### 2.8. DEGs Involved in Plant Hormone Signal Transduction

Given the functional analysis of DEGs and in light of previous studies, we then focused on those DEGs related to plant hormone signal transduction to uncover the molecular mechanisms of plant architecture in *B. napus*. Among those 96 DEGs, 80.21% of them that were associated with plant hormone signal transduction were up-regulated in dwarf plants. These were widely distributed in the signal pathways of multiple phytohormones, chiefly IAA, ABA, CK, ETH, BRs, SA (salicylic acid), and JA (Figure 6, Appendix A).

IAA and BRs exist in various plant tissues and promote internode elongation. We identified 24 and five DEGs respectively associated with IAA and BR signal transduction between the dwarf and tall plants. Sixteen and eight DEGs involved in the IAA signal transduction, these encoding auxin/indole-3-acetic acid (Aux/IAA) proteins, GH3, and auxin responsive proteins, were up- and down-regulated, respectively. In the BR signaling pathway, the accumulation of three kinase genes (*BSK*, *BAK1*, and *BIN2*) varied greatly between the tall and dwarf plants. However, between them, only two GA signaling genes were differentially expressed. In contrast to gibberellin receptor *GID1*, the expression of *GID2* encoding the F-box protein was significantly down-regulated in dwarf plants. Among the five DEGs related to CK transduction, *BnaC01g01940D*, which encodes *two-component response regulator ARR18*, had the most abundant transcripts in the samples that on average were 2.65-fold higher than in dwarf plants. Interestingly, all 16 DEGs involved in JA signal transduction were up-regulated in dwarf plants. Fourteen DEGs involved in the ABA signaling process were significantly regulated. The 12 up-regulated genes in dwarf plants are all related to the PYL-PP2C-SnRK2 complex, which plays a pivotal role in the ABA signaling process. All 11 DEGs involved in the ETH signaling pathway were up-regulated, including one ethylene-insensitive protein 3 (EIN3) gene, six ethylene-responsive transcription factor 2 (ERF2) genes, one EIN3-binding F-box (EBF1_2) gene, two ethylene receptor (ETR) genes, and one serine/threonine-protein kinase (CTR1) gene. In addition, 16 DEGs involved in the regulation of SA signaling pathway were detected, with one down-regulated and 15 up-regulated. Of the latter, transcripts of two transcription factor genes (*BnaA03g38630D* and *BnaC03g45470D*) accumulated the most in all the samples and were starkly changed between plants disparate in stature. Overall, plant hormone signal transduction is a complex process that may be central to determining plant architecture in *B. napus*.

### 2.9. Analysis of DEGs Related to Cell Wall and Cell Expansion

Seventy-six DEGs involved in cell wall structure and cell expansion displayed differing expression patterns between the tall and dwarf plants (Appendix A). In particular, 20 up-regulated and one down-regulated DEG were identified as xyloglucan endotransglucosylase/hydrolase. One (*BnaA01g04600D*) and two DEGs (*BnaA04g28830D* and *BnaC03g70880D*) that encode cellulose synthase (CesA) were respectively down- and up-regulated in dwarf plants. In addition, 52 DEGs that encode pectinesterase, polygalacturonase, pectate lyase, and endochitinase were differentially regulated in dwarf plants, in that nine of them were down-regulated and 43 up-regulated. These results indicated that genes related to cell wall and cell expansion might also participate in the formation of plant architecture in *B. napus*.

### 2.10. Analysis of DEGs Related to the Phenylpropanoid Pathway

The phenylpropanoids are precursors for lignin biosynthesis. Our KEGG analysis showed that 20 DEGs were enriched in the phenylpropanoid pathway (Appendix A). Five genes encoding cinnamyl-alcohol dehydrogenase (CAD) and caffeoyl-CoA O-methyltransferase were up-regulated in dwarf plants, while five genes encoding 4-coumarate-CoA ligase (4CL), caffeic acid 3-O-methyltransferase (COMT), and ferulate-5-hydroxylase (CYP84A) were down-regulated in them. Phenylpropanoids, as precursors for many secondary metabolites, seemed highly relevant for the phenotype difference between the tall and dwarf plants.

### 2.11. Association Analysis of BSA-Seq and Transcriptomic Data

To determine the candidate genes associated with plant architecture, we performed an association analysis by combining the BSA-Seq and transcriptomic data. A total of 542 DEGs were screened out in the five candidate regions (Appendix A). Interestingly, only four DEGs were down-regulated, whereas 99.26% of DEGs were up-regulated in dwarf plants. A KEGG pathway analysis was carried out to determine the biological roles of these 542 DEGs (Appendix A). These results show that phagosome, which plays an important role in cell growth and development, was the most enriched pathway. Six down-regulated genes (*BnaA01g02420D*, *BnaA01g02430D*, *BnaA01g07810D*, *BnaA01g08280D*, *BnaA01g00580D*, and *BnaA01g06910D*) encoding auxin responsive proteins and an *ARR18* were found in the candidate regions. Furthermore, *BnaA01g01910D*, a WUSCHEL-related homeobox gene, was down-regulated in dwarf plants. These results indicated the above genes could be the key factors underpinning the dwarf phenotype of *B. napus*.

## 3. Discussion

### 3.1. Phytohormones Might Play Major Roles in Determining Plant Architecture in B. napus

Previous studies have reported that phytohormones content has major effects on plant architecture. The altered functioning of several genes, including *BnGA2ox6*, *BnaA06.RGA*, and *BnaC07.RGA*, involved in GA biosynthesis and signaling led to a dwarf phenotype in *B. napus* [15,16,26]. In our study, the content of GAs was significantly lower in the dwarf plants than tall plants. Accordingly, a GA signaling gene, *BnaA01g13690D* (*GID2*), was down-regulated in dwarf plants. In the GA signaling process, the DELLA proteins interact with GID2 proteins, subsequently undergoing ubiquitination and degradation by binding to SCF^SLY1/GID2^ [27]. Therefore, *GID2* may play an indispensable role in the plant height trait in *B. napus*.

IAA figures prominently in many aspects of plant growth and development, including apical dominance, hypocotyl elongation, and plant height [28,29,30]. Auxin-signaling genes change their expression levels rapidly in response to a changed IAA content. In this study, the content of IAA in dwarf plants was significantly higher than that in tall plants of *B. napus*. The activity of 22 IAA signal transduction genes, including *GH3*, *AUX/IAA*, and some auxin-induced genes, were significantly changed between dwarf and tall plants. Aux/IAA proteins are known to operate as transcriptional repressors, being able to dimerize with AUXIN RESPONSE FACTOR (ARF) transcription factors to bind DNA in a rapid response to auxin [31]. The function of *GH3* in regulating plant architecture was confirmed in rice [5,6,7,28]. Our results revealed that some auxin-signaling genes were up-regulated, yet others were down-regulated in dwarf plants, implicating the importance of auxin homeostasis for plant architecture in *B. napus*.

JA is reportedly involved in the negative regulation of plant height [9]. In our study, most DEGs involved in alpha-linolenic acid metabolism, crucial for JA biosynthesis, were up-regulated in dwarf plants. Moreover, all the DEGs in the JA signaling pathway were up-regulated in dwarf plants, which implies the negative regulation of plant height by JA. These results suggested that the above DEGs are perhaps key factors in the formation of a dwarf phenotype in *B. napus*.

*CesA* and *XTHs* are cell-wall-related genes and responsible for cell division and cell expansion. Cellulose synthase is a member of glycosyltransferase family 2 and participates in the synthesis of backbones of hemicellulose [32], and *XTHs* can promote stem growth by affecting cell wall elongation in pear plants [12]. In our transcriptome data, many DEGs encoding pectinesterase, endochitinase, polygalacturonase, CesA, and XTHs were identified. Previous studies have shown that some cell-wall-related enzymes are regulated by BRs [33,34]. Although the content of BRs was similar between dwarf and tall plants, we did find several DEGs in BR signal transduction. The essential roles of BRs in plant architecture have been confirmed [35]. Three BR response genes, *BSK*, *BAK1*, and *BIN2*, underwent significantly changed expression between dwarf and tall plants. Our results show that most of the cell-wall-related genes were up-regulated in dwarf plants, but in terms of BR-responsive genes, only *BnaC01g03320D(BSK)* was significantly up-regulated. Therefore, *BnaC01g03320D* could be considered as a key gene that regulates cell-wall-related genes, such as *pectinesterase*, *endochitinase*, *polygalacturonase*, *CesA*, and *XTHs*.

We also found that the ABA content and its related genes’ expression were significantly changed between dwarf and tall plants of *B. napus*. ABA is a key stress-signaling phytohormone and plays an essential role under osmotic stress conditions [36]. Therefore, we speculate that dwarf plants may differ from tall plants in their stress tolerance, for which related phenotypes are being studied.

### 3.2. DEGs Involved in the Phenylpropanoid Pathway Are Associated with Phenotypic Differences between Tall and Dwarf Plants

During this study, we also found that the 4942C-5 has better resistance to lodging and pod shattering, traits likely related to it lignin content. Phenylpropanoids are the precursor compounds of lignin. As our results show, the DEGs between the tall and dwarf plants were significantly enriched in the phenylpropanoid pathway. In *Ricinus communis*, overexpression of *RcPAL*, a key enzyme in the phenylpropanoid pathway, caused substantial differences to its lignin content and plant height [13]. Enhancing resistance to lodging and pod shattering is always a major aim in *B. napus* breeding programs. Here, several prominent genes involved in lignin biosynthesis were identified by RNA-Seq, providing a foundation for unveiling their roles in fostering lodging and pod shattering resistance in *B. napus*.

### 3.3. WOX May Contribute to the Formation of the Dwarf Phenotype in B. napus

WUSCHEL-related homeobox proteins are critical for plant growth and development by modulating the transcription of various genes [37]. Research has shown that WOX proteins can regulate plant height via GA signaling; for example, overexpressing *OsWOX3A* in rice negatively regulated GA biosynthesis and led to a dwarf phenotype [14]; in wheat, a *TaWUS-like* gene inhibited stem elongation by affecting both the GA and BR content [38]. Furthermore, WUSCHEL can influence growth and development by suppressing ARRs, cytokinin-inducible response regulators [39]. In this study, an abnormal apical meristem was observed in dwarf plants. The expression levels of genes related to plant hormone signal transduction were significantly changed (Appendix A). Moreover, a WUSCHEL-related homeobox gene (*BnaA01g01910D*) in the candidate regions was down-regulated in dwarf plants. Hence, we speculate *BnaA01g01910D* is an important gene in the regulation of plant architecture by affecting phytohormone levels or their signal transduction in *B. napus*.

### 3.4. Chromosome Structural Variation Might Occur in 4942C-5 Plants

Gene expression and phenotypic traits might be affected by chromosome structural variation [40,41]. Microspore culture provides an efficient way to produce homozygosity-stabilized lines with mutation introduction such as DNA degradation and DNA sequence changes [42,43]. 4942C-5 with dwarf and compact plant architecture is a double haploid line obtained from microspore culture. Also, plant architecture traits of 4942C-5 are recessive traits in comparison with those of 8008. Thus, it is reasonable to speculate that certain structural variations taking the form of chromothripsis and insertional translocations might exist in 4942C-5. Evidence comes from two aspects: (1) abundant SNPs accumulated on Chromosome A01 in the dwarf pool, and candidate intervals of approximately 6.41 M on Chromosome A01 was determined by BSA-Seq (Figure 3); and (2) a total of 43.15% of the annotated genes (542) in the candidate regions were DEGs, most of which exhibited down-regulation in the candidate interval (Appendix A). Considering that sequence variations or down-regulation of key structural genes usually correlated with impairment of phytohormone biosynthetic pathway and loss-of-function mutations [44], it is plausible that DNA sequence deletions or changes exist in these candidate genes that are involved in phytohormone biosynthesis and plant growth and development, resulting in broad down-regulation of its adjacent genes and the recessive plant architecture phenotype. For future work, it would be interesting and challenging to detail these structural variations on chromosome A01 in 4942C-5.

## 4. Materials and Methods

### 4.1. Plants Materials

The *B. napus* line 4942C-5 is noted for its special plant architecture and is a double haploid line obtained from microspore culture. Specifically, a typical 4942C-5 plant exhibits a short branch length and dwarf phenotype. Reciprocal crosses between 4942C-5 and 8008, a line with normal plant architecture, were conducted to produce F_1_ progeny. These F_1_ plants with 4942C-5 as the female parent were used to generate a BC_8_ population (*n* = 965 plants) by recurrent backcrossing with 4942C-5. Plants with contrasting plant architecture, that is, one type with short branches and a dwarf phenotype and another type that is long-branched and is a tall phenotype, in the BC_8_ population were used for phytohormone profiling, and RNA-Seq and BSA-Seq analyses. For convenience, these two types of plants are designated here as dwarf and tall plants, respectively, in the following experiments.

### 4.2. Trait Evaluation

The parents and F_1_ plants were planted in a randomized complete block design with three replicate lines. Each line was planted in two rows, with a row length of 1.5 m per replicate. Phenotyping was done for 10 competitive plants and the mean of these 10 observations used as the trait value. Four traits were evaluated for parents and F_1_ plants: plant height, main inflorescence length, number of primary branches, and length of primary branches. Plant height was measured from the ground to the tip of the main shoot. Main inflorescence length was measured from the base of the last primary branch to the tip of the main shoot. Average length of primary branches is the ratio of the sum of length of all primary branches to the number of effective primary branches. All samples were stored at −80 °C for their subsequent analyses. All materials were grown at the experimental field site of Huazhong Agricultural University, China.

### 4.3. Phytohormone Profiling

A total of 36 tall and 36 dwarf individuals in the BC_8_ population were used for a phytohormone analysis, with three biological replicates (each replicate contains 12 individuals). By following a previous study [45], the phytohormones IAA, GA, ABA, SL, CK, and ETH were extracted. Briefly, the freeze-dried tissues were crushed in a mixer mill (30 Hz, 1 min). A total of 50 mg of powder was extracted with 1 mL of a methanol:water:formic acid (15:4:1, *v*/*v*/*v*) solution containing 0.001 ng of an internal standard. After vortexing, centrifuging, and concentrating it, the concentrated sample was dissolved in 100 μL of an 80% methanol/water solution. Finally, each sample was filtered through a 0.22-μm microporous membrane for the LC-MS/MS analysis.

The liquid chromatography conditions were as follows: mobile phase A, water containing 0.04% acetic acid; and mobile phase B, acetonitrile containing 0.04% acetic acid. The flow rate was set to 0.35 mL/min. The gradient program went as follows: 95% A and 5% B at 0–8 min; 5% A and 95% B at 8–9 min; and 5% A and 95% B at 9–12 min. The extraction of BRs was the same as the above methods with some modifications, which was extracted with 1 mL acetonitrile and dissolved with 100 μL of acetonitrile. The liquid chromatography conditions for BRs were as follows: mobile phase A, water containing 0.1% formic acid; and mobile phase B, acetonitrile containing 0.1% formic acid. The gradient program went as follows: 45% A and 65% B at 0–6 min; 10% A and 90% B at 6–10 min; and 45% A and 55% B at 10–12 min. The mass spectrometry conditions were as follow: ion source temperature, 550 °C; mass spectrometry, 5.5 kV and −4.5 kV; and curtain gas pressure, 35 psi. In the Q-Trap 6500+ system, each ion pair was scanned based on its optimized declustering potential and collision energy. The obtained mass spectrometry data were analyzed in Analyst v1.6.3 software. Based on the retention time (RT) and calibration curves of the standards (Appendix A), the content of phytohormones was calculated.

### 4.4. DNA Library Construction and BSA-Seq Analysis

To identify the candidate regions responsible for the dwarf phenotype of 4942C-5, from the BC_8_ population 36 tall and 40 dwarf individuals were selected. Their total genomic DNA was extracted using the Hi-DNA secure Plant Kit (Tiangen, Beijing, China). The DNA of tall and dwarf plants was mixed evenly to construct two pools, and then sequenced on an Illumina HiSeq 2000 platform. After filtering raw reads, the ensuing high-quality clean reads were aligned to the reference sequences of *B. napus* (“Darmor-bzh”, https://www.genoscope.cns.fr/brassicanapus/, last accessed date 23 February 2022) using Burrows–Wheeler Aligner software. GATK and Picard software tools were used to detect single nucleotide polymorphisms (SNPs), this entailing mark duplication, local realignment, base recalibration variant calling, and SNP filtering. The parameters of SNP-index and Δ (SNP-index) were calculated as follows: SNP-index (Tall) = MTall/(MTall + PTall), SNP-index (Dwarf) = MDwarf/(MDwarf + PDwarf), and Δ (SNP-index) = SNP-index (Dwarf) − SNP-index (Tall). The M and P parameters therein denote the sequencing depth in the mother and father lines, respectively.

### 4.5. RNA Extraction, Library Construction, Sequencing, and Bioinformatics Analysis

Total RNA was extracted from stem tips of tall and dwarf plants of the BC_8_ population at the onset of flowering. Three biological replicates were assessed, and every replicate contained 12 individuals to eliminate between-individual differences. The concentration and integrity of these RNA samples were respectively evaluated using a NanoPhotometer^®^ spectrophotometer (Thermo Fisher, Waltham, MA, USA) and an Agilent 2100 Bioanalyzer (Agilent Technologies, Santa Clara, CA, USA). These six cDNA libraries were sequenced on the Illumina HiSeq 2000 platform. After implementing strict data quality control and filtering, all clean reads were aligned to the reference sequences of *B. napus* (“Darmor-bzh”, https://www.genoscope.cns.fr/brassicanapus/, last accessed date 23 February 2022). Fragments per kilobase million reads (FPKM) were used to quantify levels of gene expression. Based on |log_2_ (pfold-change)|  ≥  1 and padj ≤ 0.05, DEGs were designated (FPKM < 1 in any samples were excluded). The K-means algorithm divides these DEGs into several sets with similar expression patterns, this done using an R (version 3.6.3) package. Functional classification and pathway analysis of the obtained DEGs were performed by Gene Ontology (GO, http://www.geneontology.org/, last accessed date 23 February 2022) and Kyoto Encyclopedia of Genes and Genomes analysis (KEGG, https://www.kegg.jp/, last accessed date 23 February 2022).

### 4.6. Quantitative Real-Time PCR (qRT-PCR) Validation of DEGs

Twelve DEGs associated with plant architecture were validated by qRT-PCR. Their cDNA was synthesized from the RNA of RNA-Seq using EasyScript^®^ One-Step gDNA Removal and cDNA Synthesis SuperMix (TRANS). All 12 specific primers were designed in Primer 3.0 software (listed in Appendix A). The qRT-PCR reactions were conducted on a CFX96 Touch Real-Time PCR machine (Bio-Rad), with three replicates, using the TransStart^®^ Green qPCR SuperMix (TRANS). The qRT-PCR parameters: 1 cycle at 95 °C for 3 min; 40 cycles of 95 °C for 10 s, 58 °C for 10 s, 72 °C for 30 s; followed by the melting curve reaction of 65 °C to 95 °C (0.1 °C/s). The actin gene (*actin7*) served as an internal reference. The 2^–ΔΔCt^ method was used to calculate relative expression levels of the 12 genes [46].

## 5. Conclusions

In this study, we characterized a *B. napus* line 4942C-5 with special plant architecture. To reveal the molecular mechanisms underlying plant architecture, genetic analysis, and phytohormone profiling, BSA-seq and RNA-seq was performed on the offspring of 4942C-5. The results showed that the dwarf and compact phenotype of 4942C-5 were recessive traits and controlled by multiple genes. Candidate regions associated with plant architecture traits were identified. We found that GA, IAA, and ABA biosynthetic and signaling pathways play important roles in affecting plant architecture. Compared to tall plants, most of the DEGs in the candidate regions were down-regulated in the dwarf plants. Seven genes involved in plant hormone signal transduction and a WUSCHEL-related homeobox gene were promising candidates responsible for the special phenotype of 4942C-5. However, functional verification of these genes was needed in the further study. Overall, 4942C-5 is a promising germplasm for breeding new rapeseed varieties with an ideal plant architecture.

## Figures and Tables

**Figure 1 ijms-23-02472-f001:**
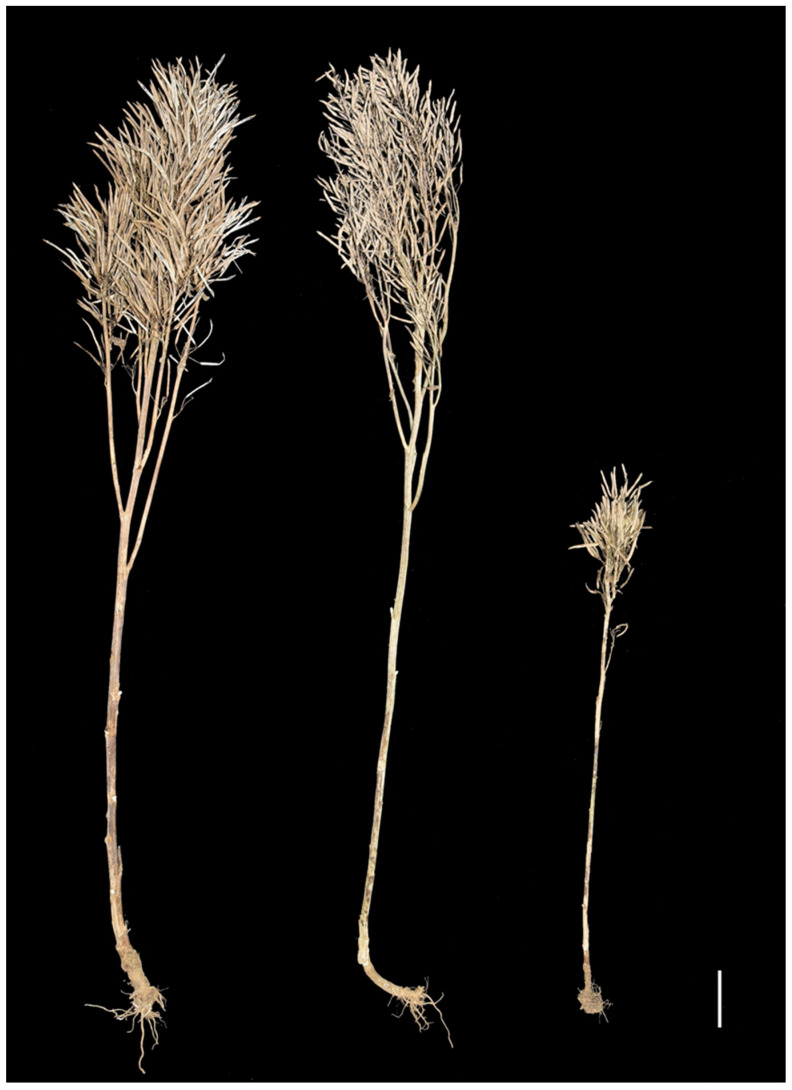
Plant architecture and height of the 8008 (**left**), 4942C-5 (**right**), and their F_1_ (**middle**) individuals at the mature stage. Scale bar = 10 cm.

**Figure 2 ijms-23-02472-f002:**
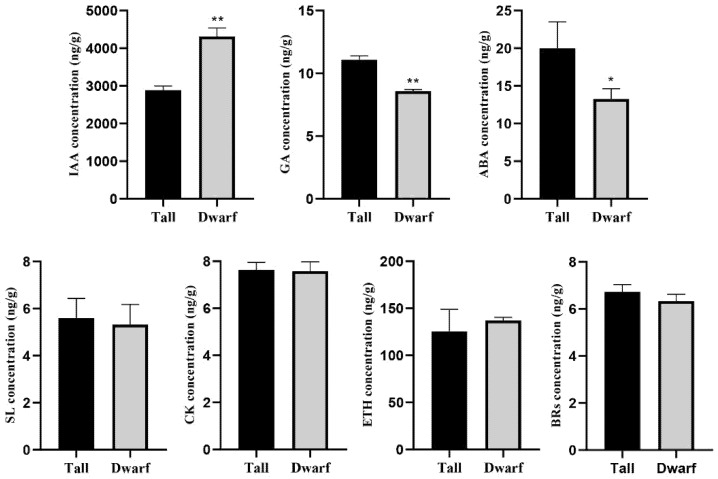
Concentrations of IAA, GA, ABA, SL, CK, ETH, and BRs between tall and dwarf plants (*Brassica napus*). Student’s *t*-test was used to compare their mean (±SE) values. Asterisks indicate significant differences: * *p* < 0.05 and ** *p* < 0.01.

**Figure 3 ijms-23-02472-f003:**
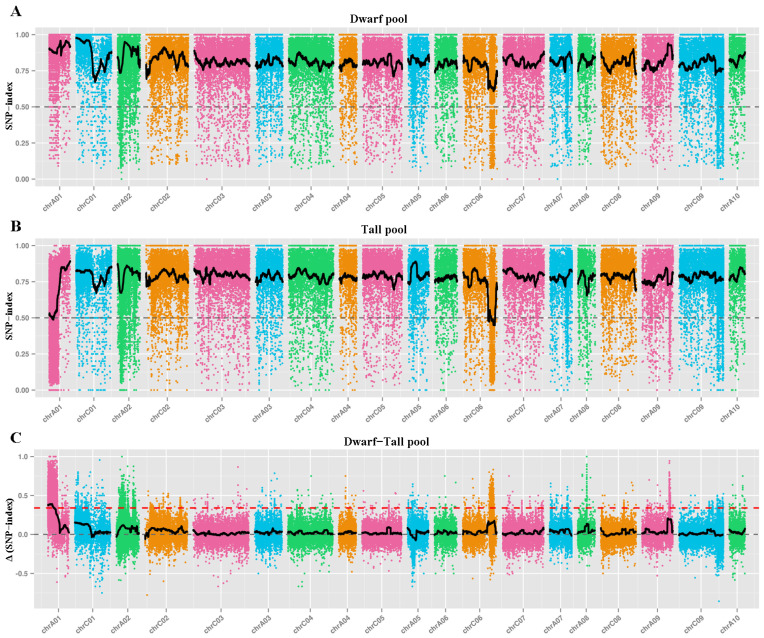
Mapping the genomic region of *Brassica napus* by BSA-seq. (**A**) The SNP index distribution of the dwarf pool from the BC8F1 population. (**B**) The SNP index distribution of the tall pool from the BC8F1 population. (**C**) Δ (SNP-index) plot of the dwarf and tall pools. The threshold line is the red dashed line.

**Figure 4 ijms-23-02472-f004:**
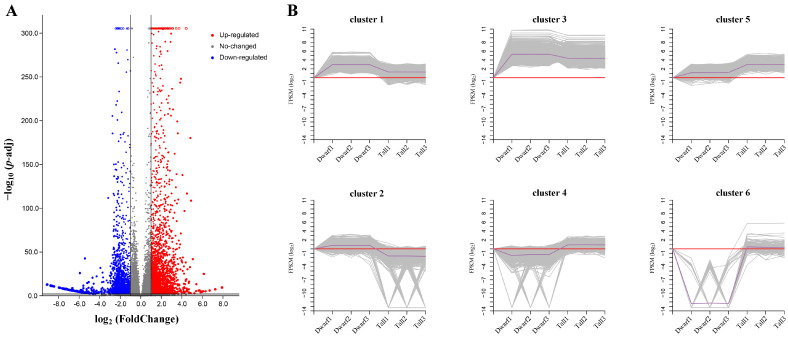
The number of DEGs (differentially expressed genes) and their expression patterns. (**A**) Scatterplots of all expressed genes between tall and dwarf plants (*Brassica napus*). Red dots are the up-regulated DEGs and blue dots are the down-regulated DEGs. (**B**) The six clusters of DEGs expression patterns. Dwarf1, Dwarf2, and Dwarf3 in each figure are three replicates, and likewise for Tall1, Tall2, and Tall3.

**Figure 5 ijms-23-02472-f005:**
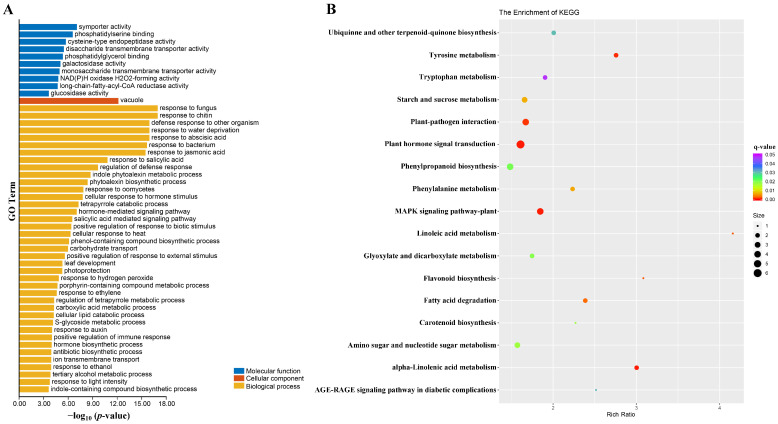
Functional analysis of DEGs (differentially expressed genes). (**A**) The top-50 enriched GO terms in molecular function, cellular component, and biological process categories. (**B**) KEGG pathway enrichment of the DEGs. The circles’ size is proportional to the number of genes, and their color denotes the range of the q-value.

**Figure 6 ijms-23-02472-f006:**
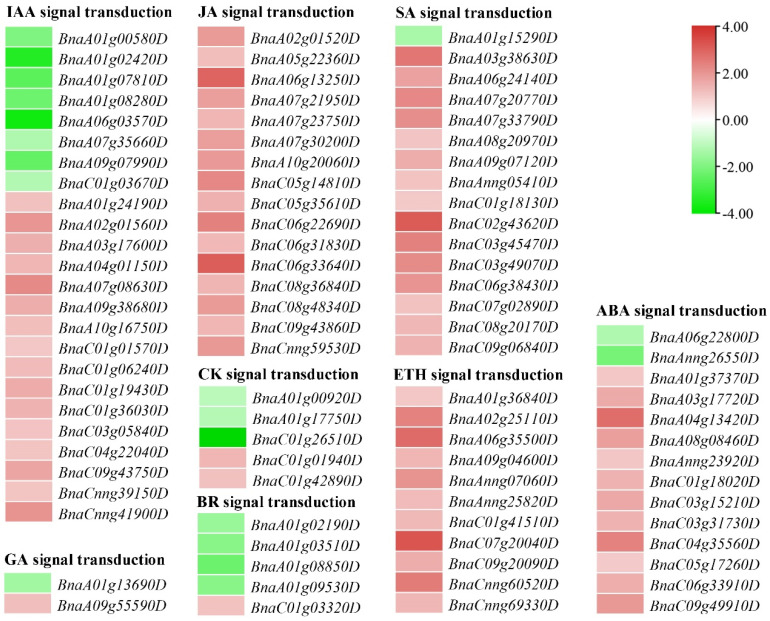
The DEGs (differentially expressed genes) for plant hormone signal transduction in the comparison of tall and dwarf plants (*Brassica napus*). Heatmaps indicate the log_2_ (fold-change) of DEGs between tall and dwarf plants, in which the red to green shading indicates log_2_ values ranging from high to low.

## Data Availability

The raw RNA-Seq data in this paper were deposited in National Center for Biotechnology Information (NCBI), under this accession number: PRJNA 792935.

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
