# Peer review of "Combined BSA-Seq Based Mapping and RNA-Seq Profiling Reveal Candidate Genes Associated with Plant Architecture in Brassica napus"

_ijms, 2022, doi:10.3390/ijms23052472_

Round 1

Reviewer 1 Report

The MS entitled  "Combined BSA-Seq based mapping and RNA-Seq profiling reveal candidate genes associated with plant architecture in Brassica napus" by Ye et al. is very interesting and well-drafted. I enjoyed reading it. Such type of approach looks promising for simple traits which show mendelian inheritance. This will reduce genotyping costs and also time. In the case of complex traits, I am sure such an approach will not work at all. Such limitations are clearly discussed here and also well understood by the community.

The use of "Plant architecture" to define dwarf and tall does not look appropriate but this is up to the Authors. Recent many papers have used such analogies. In my view, "Plant architecture" is not a single trait it is a collective appearance involving branches, branch angle, biomass, and many other component traits.

As mentioned in the first line of the Abstract - "Plant architecture is an important trait determining the yield of rapeseed (Brassica napus), for which obtaining varieties resistant to lodging and suitable for mechanical harvesting is the main goal of breeders" Authors can also demonstrate the applicability of approaches to identify loci governing lodging resistance.

On the basis of Figure 1, the traits seem to be governed by a dominant effect gene. It will be great if the author can elaborate discussion on a possible mechanism involving the identified candidate genes which can explain the observed dominant effect.

The observed difference for IAA, ABA, and GA is statistically significant but authors need to provide experiments to show its relevance in biological systems. In several instances, differences with statistical significance are not necessarily significant in real-life biology.

In Figure 3, the SNP index at significant loci on chrA01 in the dwarf pool does not look as expected. Similarly, please look at Chr07.

Dr. Humira Sonah

Author Response

Dear Madam/Sir,

The manuscript “Combined BSA-Seq based mapping and RNA-Seq profiling reveal candidate genes associated with plant architecture in Brassica napus” (ijms-1575676) has been revised with full considerations of the comments and suggestions from you and is now returned for your further reviews. It has been edited by a native English-speaker. Our main responses to the comments are attached and some changes are made in the manuscript.

Specially, we greatly appreciate for your kind and valuable comments to improve our manuscript. Thank you very much.

Best regards,

Sincerely

Jing Wen

Reviewer1:
1. The use of "Plant architecture" to define dwarf and tall does not look appropriate but this is up to the Authors. Recent many papers have used such analogies. In my view, "Plant architecture" is not a single trait it is a collective appearance involving branches, branch angle, biomass, and many other component traits.
Reply: Yes, we agree with the reviewer that plant architecture is not a single trait but the appearance of many traits. In this study, plants with contrasting plant architecture were selected for the experiments. For convenience, we designate plants with tall plant height and long branches as tall plants, and plants with dwarf and short-branched phenotype as dwarf plants. We explained it in the part of Plant materials (Line 439-444). Additionally, we reorganized the first sentence in Abstract (Line 12-15) where we mentioned that “plant architecture is an important trait” last time.

  1. As mentioned in the first line of the Abstract - "Plant architecture is an important trait determining the yield of rapeseed (Brassica napus), for which obtaining varieties resistant to lodging and suitable for mechanical harvesting is the main goal of breeders" Authors can also demonstrate the applicability of approaches to identify loci governing lodging resistance.
    Reply: It has been reported that plant height is one of the major morphological traits affecting crop resistance to lodging (Islam et al., 2007). Thus, the content of identifying loci affecting plant height and lodging resistance was added in the part of Introduction (Line 46-48), as suggested by the reviewer.

Islam MS, Peng S, Visperas RM, Ereful N, Bhuiya MSU, Julfiquar AW (2007) Lodging-related morphological traits of hybrid rice in a tropical irrigated ecosystem. Field Crop Res 101:240–248

  1. On the basis of Figure 1, the traits seem to be governed by a dominant effect gene. It will be great if the author can elaborate discussion on a possible mechanism involving the identified candidate genes which can explain the observed dominant effect.
    Reply
    : Yes, genetic analysis revealed that the plant architecture traits of 8008 was governed by dominant genes, thus the dwarf and compact phenotype of 4942C-5 were recessive traits controlled by multiple genes. As suggested, we added the discussion about the possible relationship between the candidate genes and the appearance of the recessive phenotype in the part of 3.4 (Line 410-430).

4.The observed difference for IAA, ABA, and GA is statistically significant but authors need to provide experiments to show its relevance in biological systems. In several instances, differences with statistical significance are not necessarily significant in real-life biology.
Reply: According to our results, abundant IAA accumulated in the materials and the content of IAA was 1.41- to 1.66-fold higher in dwarf than tall plants. Although there were significant differences in the content of GA and ABA between the tall and dwarf plants, the content of GA and ABA is relatively low in both tall and dwarf plants in this study. We agree with the reviewer that statistical significance is not necessarily significant in real-life biology in some cases. Thus, we need more evidence to support the significance of GA and ABA in the formation of plant architecture. A piece of evidence comes from RNA-seq analysis where GA and ABA-related DEGs were detected. And in the future, the function of candidate genes involved in GA and ABA biosynthetic and signaling pathways will be studied by transformation experiments and phytohormone spraying experiments.

  1. In Figure 3, the SNP index at significant loci on chrA01 in the dwarf pool does not look as expected. Similarly, please look at Chr07.

Reply: An SNP-index is the proportion of reads harboring the SNP that are different from the reference sequence. Considering that the dwarf pool corresponds to recessive mutants and the tall pool is the wild type, it is reasonable that SNPs accumulate somewhere such as on Chr01 in dwarf pool (Fig. 3A), while only some SNPs were detected on its counterpart in tall pool (Fig. 3B). Δ(SNP-index) was obtained by comparing the difference of SNP-index between dwarf pool and tall pool. Thus, candidate regions were identified on Chr01 but not on Chr07 or other chromosomes.

Reviewer 2 Report

The problem discussed in the manuscript is relevant, especially concerning the convenience of crop processing.
The content of the article well corresponds to the goals and objectives reflected in the introduction.
In the course of the result’s presentation, their scientific validity is presented.

However, it will be better the authors had drawn up the “Conclusion”, where they emphasized the significance of the obtained results.
The material is well presented,
The article is easy to read
Minimal comments:
Text “3.1. The 4942C-5 is germplasm for breeding new rapeseed varieties with an ideal plant architecture” better to moved from the “Discussion” section to the “Results” section
In “Materials and methods”, mention the information about the internal standards you have used.

Text “3.1. The 4942C-5 is germplasm for breeding new rapeseed varieties with an ideal plant architecture” better to moved from the “Discussion” section to the “Results” section
In “Materials and methods”, mention the information about the internal standards you have used.

Author Response

Dear Madam/Sir,

The manuscript “Combined BSA-Seq based mapping and RNA-Seq profiling reveal candidate genes associated with plant architecture in Brassica napus” (ijms-1575676) has been revised with full considerations of the comments and suggestions from you and is now returned for your further reviews. It has been edited by a native English-speaker. Our main responses to the comments are attached and some changes are made in the manuscript.

Specially, we greatly appreciate for your kind and valuable comments to improve our manuscript. Thank you very much.

Best regards,

Sincerely

Jing Wen

Reviewer2

  1. The problem discussed in the manuscript is relevant, especially concerning the convenience of crop processing. The content of the article well corresponds to the goals and objectives reflected in the introduction. In the course of the result’s presentation, their scientific validity is presented. However, it will be better the authors had drawn up the “Conclusion”, where they emphasized the significance of the obtained results.

Reply: As suggested by the reviewer, the part of Conclusion has been added in Line 542-555.

2.Text “3.1. The 4942C-5 is germplasm for breeding new rapeseed varieties with an ideal plant architecture” better to move from the “Discussion” section to the “Results” section.

Reply: Corrections have been done as suggested (Line 121-126).

3.In “Materials and methods”, mention the information about the internal standards you have used.

Reply: The information about the internal standards was presented in Table S9 (Line 480 in the part of Materials and methods).